# The Association between Ultra-Processed Foods, Quality of Life and Insomnia among Adolescent Girls in Northeastern Iran

**DOI:** 10.3390/ijerph19106338

**Published:** 2022-05-23

**Authors:** Katie Elizabeth Lane, Ian Glynn Davies, Zahra Darabi, Majid Ghayour-Mobarhan, Sayyed Saeid Khayyatzadeh, Mohsen Mazidi

**Affiliations:** 1Research Institute for Sport and Exercise Sciences, School of Sport and Exercise Sciences, Liverpool John Moores University, Liverpool L3 1AA, UK; k.e.lane@ljmu.ac.uk (K.E.L.); i.g.davies@ljmu.ac.uk (I.G.D.); 2Nutrition and Food Security Research Center, Shahid Sadoughi University of Medical Sciences, Yazd 891618-8635, Iran; z.darabi91@yahoo.com (Z.D.); khayyatzadeh@yahoo.com (S.S.K.); 3International UNESCO Center for Health Related Basic Sciences and Human Nutrition, Department of Nutrition, Faculty of Medicine, Mashhad University of Medical Sciences, Mashhad 917791-8564, Iran; ghayourm@mums.ac.ir; 4Department of Nutrition, School of Public Health, Shahid Sadoughi University of Medical Sciences, Yazd 891618-8635, Iran; 5Medical Research Council Population Health Research Unit, University of Oxford, Oxford OX3 7LF, UK; 6Clinical Trial Service Unit and Epidemiological Studies Unit (CTSU), Nuffield Department of Population Health, University of Oxford, Oxford OX3 7LF, UK; 7Department of Twin Research & Genetic Epidemiology, South Wing St Thomas’, King’s College London, London SE1 7EH, UK

**Keywords:** ultra-processed foods, insomnia, health behavior, health promotion

## Abstract

Ultra-processed foods have been associated with increased risk of chronic disease, poor overall health and psychological outcomes. This study explored the association of ultra-processed foods with quality of life in adolescent girls from northeastern Iran. In an interdisciplinary cross-sectional study, *n* = 733 adolescent girls were recruited by random cluster sampling. Assessments were completed for demographics, physical activity, anthropometric and biochemical parameters, psychological health and dietary intake. The participants were categorized into quartiles of ultra-processed food intake, and multivariable logistic regression was used in several models to investigate the association between ultra-processed food intake and psychological health. The mean age of the participants was 14 years. There were no significant differences in participant demographics for the quartiles of ultra-processed food intake including weight, waist–hip ratio, waist circumference, depression, insomnia and cardiometabolic markers related to cardiovascular disease risk. Adjusted logistic regression showed participants in the highest category of ultra-processed food consumption had an increased likelihood of reduced quality of life (OR: 1.87, 95% CI: 1.13–3.11), with a greater chance for insomnia (OR: 4.04, 95% CI: 1.83–8.94) across all models. However, no significant associations were observed between consumption of ultra-processed foods and daytime sleepiness. We highlight the association between ultra-processed food consumption and poor quality of life and insomnia amongst adolescent girls. Large longitudinal integrated public health studies in different ethnicities are needed to confirm these associations and evaluate their possible impact for optimizing health promotion programs.

## 1. Introduction

A plethora of recent evidence shows strong links between ultra-processed food consumption and the prevalence of non-communicable diseases (NCDs) such as cardiovascular disease [1,2,3], type 2 diabetes [4], certain cancers [5] and metabolic syndrome [6], all linked to obesity. Furthermore, emerging evidence shows associations of ultra-processed food consumption with childhood obesity and early-onset NCDs in children [7,8,9]. Ultra-processed foods are ready-to-eat or ready-to-heat formulations of products mostly or entirely comprising substances derived from foods that typically contain little or no whole foodstuff [10,11]. Most of these foods are considered to be inferior from a nutrition perspective, providing high levels of energy and free sugars and poor in fiber, proteins and micronutrients [12,13]. In addition to an increased risk of NCDs, ultra-processed food consumption correlates with decreased quality of life and sleep and other psychological outcomes [3]. Ultra-processed foods are defined within the NOVA classification system, which groups foods according to the extent and purpose of their industrial processing [13].

Childhood obesity is becoming increasingly common in Iran, where obesity prevalence is greater than in South East Asia, Africa and Europe [14], accounting for 8% of children under 5 years of age in 2019 [15] and 9.3% and 8.1% of adolescent boys and girls, respectively. Iranian children overconsume processed food containing high amounts of sugar and/or fats, with girls consuming higher quantities of a range of sweet and savory ultra-processed foods than boys [16].

The aim of this study was to explore the association of ultra-processed food consumption with quality of life in adolescent girls from northeastern Iran. 

## 2. Materials and Methods

### 2.1. Study Population

In this cross-sectional study, a total of 988 adolescent girls were recruited using a random cluster sampling method, from several schools in different areas in the cities of Mashhad and Sabzevar, in northeastern Iran. After random sampling was completed, the participants were invited to come to public health centers for clinical examination and anthropometric assessments. Only adolescent girls between the ages of 12 and 18 years and those without a history of chronic diseases such as colitis, diabetes, CVD, cancer and hepatitis were included. Moreover, individuals taking anti-inflammatory, antidepressant, anti-diabetic, or anti-obesity drugs as well as those on vitamin D, calcium supplementation, or hormone therapy within the last 6 months were not included. All the subjects and their parents were asked to complete a written informed consent before participating in the study. Furthermore, we excluded those who under- and over-reported their dietary intake (daily energy intake less than -3SD or more than +3SD) (*n* = 255). The study was approved by the ethic committee of Shahid Sadoughi University of Medical Sciences.

### 2.2. Demographic and Physical Activity Assessment

Demographic data, medical history, dietary supplements or drug use were collected through a standard questionnaire and by trained interviewers. Physical activity information was obtained by using the validated Modifiable Activity Questionnaire (MAQ) [17]. Physical activity levels were computed based on metabolic equivalent task (MET) hours per week, (MET-h/wk).

### 2.3. Anthropometric and Biochemical Assessment

Weight, height, waist circumferences (WC) and waist-to-height ratio (WHR) were measured by using standard protocols, and the mean of two measurements is reported. Body mass index (BMI) was calculated as weight (kg) divided by the square of the height (m^2^).

Fasting blood samples were obtained early in the morning between 8 and 10 a.m. by venipuncture of an antecubital vein after a 14 h overnight fast. The samples were collected in vacuum tubes from the subjects in a sitting position, according to a standard protocol. Serum concentrations of fasting blood glucose (FBG), triglycerides (TG), total cholesterol (TC), high-density lipoprotein–cholesterol (HDL-C), aspartate aminotransferase (AST) and alanine aminotransferase (ALT) were measured using commercial kits (Pars Azmun, Karaj, Iran) and the BT-3000 auto-analyzer (Biotechnica, Rome, Italy). LDL-C was calculated using Friedewald formula if the serum concentration of TG was lower than 4.52 mmol/L.

### 2.4. Dietary Assessment

Assessment of dietary intakes was performed by a qualified dietitian using a validated food frequency questionnaire (FFQ) containing 147 food items [18]. The frequency of food items intake during the last year was evaluated by asking the participants about their daily, weekly, monthly and yearly intake. The conversion of the food portion sizes to grams was performed by household measures. We used the Nutritionist IV software (version 7.0; N4 Computing, Salem, OR, USA), modified for Iranian foods, to analyze the dietary intakes. Ultra-processed food was defined by total intake of mass-produced packaged breads, sausages and other reconstituted meat products, confectionary, ice cream, ice pops and frozen yogurts, biscuits, pastries, buns, cakes, industrial French fries, sugar-sweetened milk-based drinks, sauces, dressing and gravies, fruit drinks and iced teas, carbonated soft drinks, packaged salty snacks, frozen pizza, margarine and other spreads identified as ultra-processed foods in the Australian processed food classification system [19].

### 2.5. Assessment of Psychological Health

The Insomnia Severity Index (ISI) questionnaire was used to evaluate insomnia [20]. The ISI questionnaire has seven questions and evaluates the answers with a score in the range between 0 and 4, that stratify insomnia into four categories as follows: 0 (None), 1 (Mild), 2 (Moderate), 3 (Severe) and 4 (Very Severe). The total score of ISI ranges from 0 to 28 points. Insomnia was defined if the total ISI score was >7.

A Persian translation of the Epworth Sleepiness Scale (ESS) was used for the assessment of daytime sleepiness, and a confirmation of its reliability and validity has been published previously [21]. This questionnaire asks the respondents to rate their sleepiness in eight daily situations from 0 to 3, giving a total score of 0 (no daytime sleepiness) to 24 (the highest degree of daytime sleepiness). EDS was defined as an ESS score ≥10 [22]. 

To assess health-related quality of life, the SF-12v2 questionnaire was used. This questionnaire is a short form of the SF-36 questionnaire and an improved version of SF-12v1 [23]. The validity and reliability of the questionnaire were confirmed in Iran [24]. The questionnaire has 12 items evaluating 8 domains of health including physical functioning, role limitations because of physical problems, bodily pain, general health, vitality, social functioning, role limitations because of emotional problems, and mental health. The range of quality of life scores is between 0 (the worst quality of life) and 100 (the best quality of life). The median quality of life score is 43. The subjects were considered as having a poor quality of life if their scores were lower than 43.

### 2.6. Statistical Analysis

The participants were categorized by quartiles of ultra-processed food intake. The normality of the data was assessed using the Kolmogorov–Smirnov test. To compare the general characteristics of the study participants across quartiles of ultra-processed food intake, one-way ANOVA was used. Linear regression analysis was performed to explore differences for dietary intakes of ultra-processed food components between quartiles of ultra-processed food intake. The association between quartiles of ultra-processed food intake with insomnia, poor quality of life and daytime sleepiness was evaluated using a multivariate regression in the crude and adjusted models. Age and energy intake were adjusted in Model I. Additionally, BMI percentile was adjusted in Model II. A final adjustment was done for physical activity. Statistical analyses were conducted using the statistical package for social sciences (SPSS), version 23. A *p*-value < 0.05 was considered significant.

## 3. Results

### 3.1. General Characteristics of the Study Participants

The general characteristics of the study participants across categories of ultra-processed food intake are shown in Table 1. There were no significant differences for age, BMI percentile, WC, WHR, physical activity, scores of insomnia, daytime sleepiness and quality of life, FBG, TG, TC, HDL-C, AST and ALT between quartiles of ultra-processed food intake.

### 3.2. Dietary Intakes of the Study Participants

The dietary intakes of ultra-processed food components across quartiles of ultra-processed food intake are provided in Table 2. The participants in the fourth quartiles of ultra-processed food intake consumed significantly higher amounts of mass-produced packaged breads, sausages and other reconstituted meat products, confectionary, ice cream, ice pops and frozen yogurts, biscuits, pastries, buns and cakes, industrial French fries, sweetened milk-based drinks, sauces, dressing and gravies, fruit drinks and iced teas, carbonated soft drinks, packaged salty snacks, and frozen pizza.

### 3.3. The Association between Ultra-Processed Food Intake and Insomnia, Daytime Sleepiness and Poor Quality of Life

The crude and adjusted associations between ultra-processed food intake and insomnia, daytime sleepiness and poor quality of life are indicated in Table 3. The participants in the highest quartile of ultra-processed food intake compared with those in the lowest quartile had 2.77 higher odds of insomnia (OR: 2.77; 95% CI: 1.5–5.10, *p* <0.01). This association remained significant after adjustments in several adjusted models [(Model I; OR: 3.90; 95% CI: 1.81–8.77, *p* <0.01), (Model II; OR: 4.04; 95% CI: 1.83–8.92, *p* <0.01), (Model III; OR: 4.04; 95% CI: 1.83–8.94, *p* <0.01)]. Although no significant relation was observed between ultra-processed food intake and poor quality of life in the crude model, greater adherence to ultra-processed food intake was associated with higher odds of poor quality of life in the adjusted models [(Model I; OR: 1.87; 95% CI: 1.13–3.10, *p* = 0.01), (Model II; OR: 1.83; 95% CI: 1.10–3.04, *p* = 0.01), (Model III; OR: 1.87; 95% CI: 1.13–3.11, *p* = 0.01)]. There was no significant association between ultra-processed food intake and daytime sleepiness in both crude and adjusted models.

## 4. Discussion

Emerging evidence of the detrimental impact of child and adolescent diets with a high content of ultra-processed food is highlighting an important aspect of public health strategies [25,26]. Furthermore, links have been established between increased ultra-processed food consumption, early onset of NCDs and decreased quality of life in children and adolescents [7,9,14,27]. Childhood obesity is becoming increasing common in Iran where children are overconsuming processed food containing high amounts of sugar and/or fats [16]. This study used an interdisciplinary, cross-sectional approach to explore the association of ultra-processed foods with quality of life in adolescent girls from northeastern Iran.

### 4.1. The Impact of Ultra-Processed Food Intake on General Health Characteristics

One-way ANOVA showed no significant differences for age, BMI percentile, WC, WHR, physical activity, scores of insomnia, daytime sleepiness and quality of life, FBG, TG, TC, HDL-C, AST and ALT between quartiles of ultra-processed food intake in the study participants. Our findings are in partial agreement with a systematic review and meta-analysis of 43 observational studies by Lane et al. [9], where cardiometabolic risk parameters such as overweight, obesity, abdominal obesity and metabolic diseases were shown to be associated with the consumption of ultra-processed food in adults; however, associations in children and adolescents were less clear. These findings suggest the frequent consumption of ultra-processed food over time may lead to negative effects on subclinical cardiovascular disease markers that may be more apparent later in life, for example in early adulthood. Further longitudinal follow-up studies are needed to confirm these associations.

### 4.2. The Impact of Ultra-Processed Foods on Quality of Life

Adjusted logistic regression showed that the participants in the highest category of ultra-processed food consumption had an increased likelihood of reduced quality of life (OR: 1.87, 95% CI: 1.13–3.11,) across all models in our study. Our findings relating to the impact of ultra-processed food consumption on quality of life are novel to Iran and are in agreement with recent literature focusing on other countries. da Costa et al. [28] evaluated the association between lifestyle behaviors and quality of life in 739 Brazilian adolescents. The processed food score was unfavorably associated with participant reported health-related quality of life (95%CI: −0.78; −0.13, *p* < 0.05). In a systematic review and meta-analysis of 17 studies (47,932 participants), Wu et al. [29] evaluated the influence of diet quality and dietary behavior on health-related quality of life in children and adolescents in the general population. For the 10 studies included in the meta-analysis, children with an unhealthy diet had a significantly lower health-related quality of life than children with a healthy diet (pooled mean difference = 3.45, 95% CI 2.40, 4.50, *p* < 0.0001). However, diet quality was assessed using a number of methods and variables, which did not specifically measure total ultra-processed food consumption. For the main part, unhealthy diets were defined as those with a high content of fatty foods, sweets and salty snacks [29], in line with other studies that evaluated processed and ultra-processed food intake in this age group. Further evidence showed links between ultra-processed food intake and excessive free sugar intake. Babashahi et al. [16] showed processed sugar and sweets, followed by oils, biscuits and cakes, were the most commonly consumed processed food groups in a systematic review and meta-analysis of 10 studies on 67,093 Iranian children aged 4–12 years. Other countries have shown similar results; for example, Machado et al. [30] showed that ultra-processed food consumption drives an excessive free sugar intake among all age groups in Australia. Older children and adolescents were the highest consumers (53.1% and 54.3% of total energy, respectively) of processed foods, and a statistically significant linear association was found between quintiles of ultra-processed food consumption and both the average intake of free sugars (*p* < 0.001) and the prevalence of excessive free sugar intake (*p* < 0.001).

Emerging evidence suggests interventions to reduce ultra-processed food consumption in this age group offer the opportunity to improve the quality of life. Poll et al. [27], showed reductions in the consumption of ultra-processed foods resulted in (non-significantly) decreased body mass index and waist circumference and a tendency towards increased quality of life in 62 Brazilian adolescents with overweight or obesity during a 6-month nutrition education intervention. However, larger sample sizes and longer follow-up periods are needed to ratify these results.

### 4.3. The Impact of Ultra-Processed Foods on Sleep Quality

Adjusted logistic regression showed participants in the highest category of ultra-processed food consumption had a greater chance of insomnia (OR: 4.04, 95% CI: 1.83–8.94). Our findings are in agreement with a number of studies in different locations. Sousa et al. [31] showed those with a higher energy intake from ultra-processed foods and a lower energy intake from fresh minimally processed foods had higher prevalence of poor sleep quality in 2499 older Brazilian adolescents (aged 18–19 years). The majority of participants (57.1%) reported having poor sleep quality; however, the fourth quartile of ultra-processed food consumption (44.3%–81.8% of total calories; PR = 1.14; 95% CI: 1.03, 1.27) was associated with a higher prevalence of poor sleep quality. A study of 509 Italian children/adolescents showed the overall energy intake from ultra-processed foods (defined using NOVA classification) was 25·9% [95% CI (24·8, 27·0); IQR: 17·0–34·1]. Increased ultra-processed food consumption showed direct links to low sleep quality [β = 2·34; 95% CI (1·45, 3·23)], with boys consuming 2 percent points more ultra-processed foods in total energy than girls [β = 2·01; 95% CI (0·20, 3·82)] [32].

There are limitations to the present study. Firstly, this was a cross-sectional study; therefore, it is fundamentally difficult to determine whether or not the observed relationships are causal. Secondly, data provided on ultra-processed food consumption and quality of life used self-reported questionnaires, and lifestyle choices and lived experiences were not considered. Finally, sleep quality can be influenced by other factors such as psychological outcomes that were not addressed in our statistical model. It should also be considered that the study participants were selected from the cities of Mashhad and Sabzevar, in northeastern Iran, and a generalization of the findings should be carried out with caution. In addition, ultra-processed food consumption was evaluated using the Australian processed food classification system [33], in which foods are categorized based on the level of food processing under the NOVA classification system [10,13]. Whilst the NOVA classification system is widely used in the literature [1,2,4,5], its use may also represent a limitation. There is a lack of consensus over its use instead of traditional epidemiologic approaches that rely on forging links between nutrient intakes and chronic diseases, with the subsequent identification of foods that merit consideration in public health nutrition strategies [12,34]. In consideration of the emerging relationship between ultra-processed foods and the risk of NCDs, the Australian processed food classification system represents a useful dietary assessment tool to establish both the level of processing and the opportunity to extend knowledge on diet quality and dietary patterns. However, the use of an Australian classification system in the present study may present a limitation, as it might not translate to specific versions of ultra-processed foods unique to the Iranian culture. Babashahi et al. [16] showed Iranian children have high consumption of miscellaneous food groups that include foods high in fats and sugars that do not appear as processed foods in the Iranian food pyramid.

## 5. Conclusions

In conclusion, the present study highlighted associations between ultra-processed food consumption and quality of life in 733 adolescent girls from northeastern Iran. There were no significant differences in participant demographics for the quartiles of ultra-processed food intake including weight, waist–hip ratio, waist circumference, depression, insomnia and cardiometabolic markers related to cardiovascular disease risk. We established that participants in the highest category of ultra-processed food consumption have an increased likelihood for reduced quality of life and a greater chance of insomnia. These findings are in agreement with those from other countries, where studies have shown similar associations. No significant associations were observed between consumption of ultra-processed foods and daytime sleepiness. Furthermore, large longitudinal integrated public health studies in different ethnicities are needed to confirm these associations and inform intervention studies for health promotion.

## Figures and Tables

**Table 1 ijerph-19-06338-t001:** General characteristics of the study participants by quartile of ultra-processed food intake.

	Q1	Q2	Q3	Q4	*p*-Value *
Age (year)	14.51 ± 1.52	14.51 ± 1.57	14.53 ± 1.49	14.49 ± 1.55	0.99
Height (cm)	156.97 ± 5.92	156.89 ± 6.79	158.20 ± 6.01	157.58 ± 5.65	0.14
Percentile BMI (kg/m^2^)	51.60 ± 29.85	47.39 ± 27.93	45.05 ± 28.26	47.30 ± 29.43	0.17
Weight (kg)	53.92 ± 13.53	51.90 ± 10.50	52.44 ± 11.13	52.73 ± 11.78	0.41
Waist circumference (cm)	71.25 ± 9.99	69.96 ± 8.30	70.72 ± 8.68	70.08 ± 9.33	0.49
Waist-to-hip ratio	0.76 ± 0.06	0.76 ± 0.05	0.77 ± 0.07	0.76 ± 0.05	0.39
Metabolic equivalent for task (h/week)	45.09 ± 2.81	5.11 ± 3.26	54.60 ± 3.68	45.70 ± 3.91	0.19
Depression score	9.89 ± 9.04	11.23 ± 9.89	11.81 ± 9.31	10.75 ± 8.79	0.23
Insomnia score	6.55 ± 5.05	8.16 ± 6.25	7.80 ± 5.89	8.76 ± 5.28	0.06
Aggression score	78.96 ± 19.29	76.53 ± 19.95	77.48 ± 21.25	80.98 ± 20.81	0.17
Daytime sleepiness score	6.60 ± 3.86	7.41 ± 3.78	7.32 ± 3.60	7.07 ± 3.84	0.2
Quality of life score	42.67 ± 8.09	42.71 ± 8.02	41.34 ± 8.22	41.57 ± 7.54	0.22
Total cholesterol (mg)	163.22 ± 25.28	158.23 ± 31.61	162.56 ± 27.64	160.00 ± 28.49	0.37
Triglyceride (mg)	83.68 ± 38.61	84.37 ± 42.28	89.40 ± 40.45	83.00 ± 37.80	0.43
Low-density lipoprotein (mg)	100.61 ± 23.95	97.29 ± 26.84	102.88 ± 25.74	98.36 ± 25.71	0.2
High-density lipoprotein (mg)	47.67 ± 9.06	46.18 ± 8.42	47.43 ± 8.04	46.68 ± 9.00	0.38
Fast blood sugar (mg)	86.78 ± 10.77	85.30 ± 12.27	86.93 ± 12.66	85.88 ± 10.57	0.53
Aspartate aminotransferase (IU/L)	19.71 ± 5.67	20.09 ± 6.92	20.18 ± 6.58	19.31 ± 5.43	0.55
Alanine aminotransferase (IU/L)	11.74 ± 5.50	11.50 ± 7.15	12.08 ± 7.94	11.20 ± 6.18	0.68
Gamma-glutamyltransferase (IU/L)	14.04 ± 10.17	11.64 ± 7.13	12.54 ± 6.68	12.21 ± 9.19	0.21

BMI: body mass index. Values are means ± SDs. * Obtained from one-way ANOVA for continuous variables and chi-squared test for categorical variables.

**Table 2 ijerph-19-06338-t002:** Dietary intake of the study participants by quartile of ultra-processed food intake.

	Q1 (*n* = 183)	Q2 (*n* = 183)	Q3 (*n* = 184)	Q4 (*n* = 183)	*p* Trend *
Energy (kcal)	2049.95 ± 669.83	2584.21 ± 680.84	2897.45 ± 671.77	3355.68 ± 702.49	<0.001
Mass-produced packaged breads (gr)	2.06 ± 4.45	5.63 ± 14.44	9.30 ± 19.30	16.06 ± 32.89	<0.001
Sausages and other reconstituted meat products (gr)	2.966 ± 3.94	5.98 ± 7.88	9.84 ± 12.196	11.93 ± 11.39	<0.001
Confectionary (gr)	13.306 ± 11.26	18.75 ± 14.56	25.30 ± 19.30	32.98 ± 24.39	<0.001
Ice cream, ice pops and frozen Yogurts (gr)	5.326 ± 7.38	9.81 ± 12.86	13.84 ± 17.47	23.60 ± 47.13	<0.001
Biscuits (gr)	2.536 ± 5.07	3.83 ± 8.48	5.23 ± 9.05	7.77 ± 14.90	<0.001
Pastries, buns and cakes (gr)	8.766 ± 11.47	16.32 ± 15.74	24.41 ± 22.63	28.76 ± 36.88	<0.001
Industrial French fries (gr)	4.516 ± 5.70	11.94 ± 11.63	20.78 ± 23.33	33.10 ± 34.88	<0.001
Margarine and other spreads (gr)	0.05 ± 0.38	0.37 ± 2.36	0.71 ± 4.56	0.45 ± 2.64	0.99
Sugar-sweetened milk-based drinks (gr)	13.92 ± 35.22	18.85 ± 35.52	30.48 ± 53.22	38.13 ± 64.25	<0.001
Sauces, dressing and gravies (gr)	2.84 ± 3.66	5.51 ± 6.41	7.98 ± 14.69	10.20 ± 11.26	<0.001
Fruit drinks and iced teas (gr)	4.33 ± 7.42	11.72 ± 16.16	19.35 ± 26.68	62.30 ± 87.62	<0.001
Soft drinks, carbonated (gr)	7.08 ± 10.32	24.49 ± 24.87	42.14 ± 42.18	157.49 ± 167.96	<0.001
Packaged salty snacks (gr)	2.10 ± 2.71	4.13 ± 5.15	6.47 ± 6.67	10.13 ± 11.97	<0.001
Frozen pizza (gr)	7.646 ± 34.27	7.66 ± 15.12	10.86 ± 20.01	15.33 ± 23.36	<0.01

Values are means ± SDs. * Obtained from linear regression.

**Table 3 ijerph-19-06338-t003:** Multivariable-adjusted odds ratio of the associations between ultra-process food intake and insomnia, depression, aggression, poor quality of life, daytime sleepiness.

	Q1	Q2	Q3	Q4	*p*-Value ^1^	*p* Trend
Insomnia						
Crude	1.00	1.65(0.90–3.02)	1.93(1.06–3.51)	2.77(1.50–5.10)	<0.01	<0.01
Model 1	1.00	1.92(1.00–3.67)	2.41(1.23–4.74)	3.90(1.81–8.77)	<0.01	<0.01
Model 2	1.00	1.94(1.01–3.72)	2.45(1.24–4.85)	4.04(1.83–8.92)	<0.01	<0.01
Model 3	1.00	1.96(1.02–3.76)	2.46(1.24–4.86)	4.04(1.83–8.94)	<0.01	<0.01
Depression						
Crude	1.00	1.43(0.88–2.32)	1.58(0.98–2.55)	1.06(0.64–1.76)	0.79	0.7
Model 1	1.00	1.54(0.94–2.55)	1.80(1.06–3.04)	1.30(0.71–2.36)	0.38	0.32
Model 2	1.00	1.53(0.93–2.53)	1.77(1.05–2.00)	1.28(0.70–2.33)	0.41	0.35
Model 3	1.00	1.53(0.93–2.53)	1.74(1.04–2.99)	1.27(0.70–2.33)	0.42	0.37
Aggression						
Crude	1.00	0.82(0.54–1.23)	0.79(0.52–1.19)	0.95(0.63–1.44)	0.34	0.8
Model 1	1.00	0.80(0.52–1.22)	0.76(0.48–1.19)	0.90(0.54–1.48)	0.68	0.66
Model 2	1.00	0.80(0.52–1.22)	0.76(0.48–1.99)	0.90(0.54–1.48)	0.67	0.66
Model 3	1.00	0.80(0.52–1.23)	0.76(0.47–1.17)	0.87(0.53–1.44)	0.6	0.57
Poor quality of life						
Crude	1.00	1.14(0.75–1.72)	1.40(0.93–2.12)	1.43(0.95–2.17)	0.08	0.05
Model 1	1.00	1.26(0.82–1.93)	1.67(1.06–2.63)	1.87(1.13–3.10)	0.01	<0.01
Model 2	1.00	1.23(0.80–1.90)	1.62(1.03–2.57)	1.83(1.10–3.04)	0.01	0.01
Model 3	1.00	1.23(0.80–1.90)	1.66(1.05–2.62)	1.87(1.13–3.11)	0.01	<0.01
Daytime sleepiness						
Crude	1.00	1.33(0.82–2.17)	1.01(0.61–1.66)	0.90(0.54–1.49)	0.68	0.44
Model 1	1.00	1.36(0.82–2.26)	1.04(0.60–1.80)	0.95(0.51–1.75)	0.88	0.63
Model 2	1.00	1.36(0.82–1.26)	1.04(0.60–1.80)	0.95(0.51–1.76)	0.88	0.63
Model 3	1.00	1.36(0.82–1.26)	1.04(0.60–1.80)	0.95(0.51–1.76)	0.88	0.63

^1^ Last quartile compared to first quartile. Model 1: Adjusted for age and energy intake. Model 2: Additionally adjusted for BMI percentile. Model 3: Additionally adjusted for physical activity.

## Data Availability

Not applicable.

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
