# Peer review of "The Association between Ultra-Processed Foods, Quality of Life and Insomnia among Adolescent Girls in Northeastern Iran"

_ijerph, 2022, doi:10.3390/ijerph19106338_

Round 1

Reviewer 1 Report

  1. The study hypothesis is clear in the title of the manuscript " ... with chance ...", which means it may not affect insomnia and quality of life.
  2. The accusation of ultra-processed foods was based on their high salt, high fat, high sugar, and containing little natural food materials. In the study, no direct cause and effect to prove the accusation.
  3. Regarding the diet of subjects, it said that FFQ containing 147 food items was used. Subjects were instructed to answer the frequency of food items intake during the last year on a daily, weekly, monthly, and yearly basis. The accuracy of the data obtained in Table 2 would be questioned.
  4. What basis of the ultra-processed food intake presented in Table 2 was?
  5. Why the insomnia of adolescent girls would be related to the diet? Would it be more logical to investigate their lifestyle, e.g., what are the most things doing at daytime or nighttime? Any trouble they have in family or school?
  6. What were the Model I and Model II? Statistically, how factors, e.g., physical activity level, age, and energy intake, were adjusted in the analysis?
  7. P-value < 0.05 was considered significant. What is the value of 0.05?
  8. According to data In Table 1, what would be the height of subjects in different quartiles? 
  9. In line 73, data (daily energy intake) over or below 3SD subjectively excluded was not well explained. The daily energy intake data was not seen in Table 2.
  10. How did we look at the higher SD of data in Table 2?

Author Response

Reviewer 1

  1. The study hypothesis is clear in the title of the manuscript " ... with chance ...", which means it may not affect insomnia and quality of life.

Reply:       We have amended the title to reflect this feedback

  1. The accusation of ultra-processed foods was based on their high salt, high fat, high sugar, and containing little natural food materials. In the study, no direct cause and effect to prove the accusation.

Reply:       Thank you for pointing this out, we have outlined this as a study limitation, please see lines 267 – 268.

  1. Regarding the diet of subjects, it said that FFQ containing 147 food items was used. Subjects were instructed to answer the frequency of food items intake during the last year on a daily, weekly, monthly, and yearly basis. The accuracy of the data obtained in Table 2 would be questioned.

Reply: A range of different tools, from 24-h recalls or dietary diaries to food frequency questionnaires (FFQ), are used nowadays to measure dietary intakes (Willett, 1998, Shim et al., 2014). FFQs are the most commonly employed tools in studies of nutritional epidemiology because they are inexpensive, easy to develop and provide useful information on dietary intake over a long period of time. Furthermore, this tool can be used to conduct personal interviews. This is extremely important when studying population groups with high illiteracy rates. The FFQ should be developed specifically for the target population of the study since diet may be influenced by ethnicity, culture, economic status and environmental factors. The FFQ used in our study is a Persian validated and reliable questionnaire that applied in several previous studies (Mirmiran et al., 2010). Also, ultra-processed food intake among different population was assessed by FFQ (Khandpur et al., 2021, Lauria et al., 2021) (Mendonça et al., 2016).

  1. What basis of the ultra-processed food intake presented in Table 2 was?

Reply: Food items and/or the underlying ingredients were classified according to NOVA food classification system into the following four groups: Group 1 -Unprocessed or minimally processed foods, Group 2 - Processed culinary ingredients, Group 3 - Processed foods Group 4 - Ultra-processed foods. NOVA is now recognised as a valid tool for nutrition and public health research, policy and action, in reports from the Food and Agriculture Organization of the United Nations and the Pan American Health Organization. The fourth NOVA group is of ultra-processed food and drink products. These are industrial formulations typically with five or more and usually many ingredients. Such ingredients often include those also used in processed foods, such as sugar, oils, fats, salt, anti-oxidants, stabilisers, and preservatives (Machado et al., 2017, Monteiro et al., 2016).

  1. Why the insomnia of adolescent girls would be related to the diet? Would it be more logical to investigate their lifestyle, e.g., what are the most things doing at daytime or night time? Any trouble they have in family or school?

Reply: Thank you, this would indeed make an interesting study, however this type of data was not collected in the present study. We will of course consider a lifestyle evaluation as future research. We have further explanation of this to the study limitations (lines 269 – 270)

  1. What were the Model I and Model II? Statistically, how factors, e.g., physical activity level, age, and energy intake, were adjusted in the analysis?

Reply: In this study, physical activity level, age, energy intake and BMI percentile were considered as confounding factor, because they can cause biased estimates of associations between ultra-process food intake and psychological disorder. Adjusting variables enhances the internal validity of a study by limiting the influence of confounding and other extraneous variables. This helps you establish a correlational or causal relationship between your variables of interest. Analytic methods of adjustment attempt to determine how the groups would have compared if they had been comparable with respect to one or more confounding factors. As such, they provide an estimate of association that is closer to the truth. Logistic regression permits the use of continuous or categorical predictors and provides the ability to adjust for multiple predictors. This makes logistic regression especially useful for analysis of observational data when adjustment is needed to reduce the potential bias resulting from differences in the groups being compared (AC Sterne and R Kirkwood, 2003).

  1. P-value < 0.05 was considered significant. What is the value of 0.05?

Reply: The p-value is a number, calculated from a statistical test, that describes how likely you are to have found a particular set of observations if the null hypothesis were true. P-values are used in hypothesis testing to help decide whether to reject the null hypothesis. The smaller the p-value, the more likely you are to reject the null hypothesis. Statistical significance is another way of saying that the p-value of a statistical test is small enough to reject the null hypothesis of the test. The most common threshold is p < 0.05; that is, when you would expect to find a test statistic as extreme as the one calculated by your test only 5% of the time.

  1. According to data in Table 1, what would be the height of subjects in different quartiles?

Reply: The height of subjects in different quartiles was added in table1.

  1. In line 73, data (daily energy intake) over or below 3SD subjectively excluded was not well explained. The daily energy intake data was not seen in Table 2.

Reply: Over or under reporting is a major concern in dietary assessment. Energy underreporting is considered to be present when reported intakes are substantially lower than true energy intakes. Energy over reporting is considered to be present when reported intakes are substantially higher than true energy intakes.  In all observational nutrition studies, the participants which over or under reported energy intake (estimated by ffq) are excluded from studies. We can remove these individuals by calculating upper boundary and lower boundary by taking 3 standard deviation from the mean of the values (>+3SD or <-3SD). Also. Daily energy intake data was added to table2.

  1. How did we look at the higher SD of data in Table 2?

Reply: dietary intakes of study participants has been not normally distributed. Therefore, the standard deviation is large and close to mean. However, this is common in nutrition studies and has been reported in previous data (Khodarahmi et al., 2016, Vasmehjani et al., 2021). However, we provided Median (interquartile range)

P-value*

Q4

(N=183)

Q3

(N=184)

Q2

(N=183)

Q1

(N=183)

<0.001

3.26 (0-16.3)

0 (0-9.80)

0 (0-6.53)

0 (0-3.26)

Mass-produced packaged breads (gr)

<0.001

8.3 (3.20-18.60)

4.90 (2.04-11.81)

4.46 (0.80-7.10)

1.30 (0.10-4.66)

Sausage and other reconstituted meat products(gr)

<0.001

28.70 (15.47-42.90)

11.34 (20.60-35.87)

8.60 (15.20-25.30)

5.01 (9.80-19.20)

Confectionary(gr)

<0.001

3.49 (10.51-23.40)

2.72 (7.42-20.09)

2.18 (5.25-12.10)

0.90 (2.50-5.90)

Ice cream, ice pops and frozen yogurts(gr)

<0.001

2.30 (0.30-8.00)

2.30 (0.30-7.17)

1.30(0.30-2.42)

1.10 (0.10-2.33)

Biscuits(gr)

<0.001

21.40 (7.10-50.00)

21.40 (7.10-48.22)

14.20 (3.30-21.40)

5.00 (1.66-14.20)

Pastries, buns, and cakes(gr)

<0.001

15.70 (7.33-7.10)

15.70 (3.70-31.40)

7.33 (3.66-15.70)

3.66 (0.30-7.30)

Industrial French fries(gr)

0.99

16.10 (0-45.00)

22.50 (0.60-45.00)

6.40 (0-22.50)

0.10 (0-12.90)

Margarine and other spreads(gr)

<0.001

16.60 (0.70-35.70)

8.30 (0-35.70)

8.30(0-25.00)

0.70 (0-16.70)

Milk-based drinks(gr)

<0.001

2.74(6.42-13.30)

2.14 (4.55-10.82)

1.20(3.30-7.00)

0.50 (1.40-4.30)

Sauces, dressing and gravies(gr)

<0.001

28.60 (6.70-85.70)

13.30 (0-28.50)

6.66 (0-13.30)

0.54 (0 -6.70)

Fruit drinks and iced teas(gr)

<0.001

114.00 (57.10-200.00)

28.60 (6.70-57.10)

13.30 (6.66-28.60)

3.80 (0 -6.70)

Soft drinks, carbonated(gr)

<0.001

5.00 (2.30-15.00)

5.00 (1.50-10.00)

2.33(1.16-5.00)

1.20 (0.10-2.33)

Packaged salty snacks(gr)

<0.01

6.20 (1.20-15.00)

2.50 (0-15.00)

1.23 (0-15.00)

1.20 (0 -6.16)

Frozen pizza(gr)

Values are as median [IQR, 25th-75th percentile]

*Obtained from Kruskal-Wallis Test

Reviewer 2 Report

Need to detail in abstract factors that were NOT significantly different such as weight, cholesterol levels, waist to hip ratio, metabolic task, waist circumference, depression AND insomnia – emphasizing only minor differences in measures does not accurately reflect the data

Definition of "ultra-processed" needs explaining based on the literature not just a list of foods.

There is debate in the literature that needs to be acknowledge and addressed the only citation is Monteiro et al. The lack of consensus around the NOVA system should be noted see Gibney, MJ, Forde, CG, Mullally, D et al. (2017) Ultra-processed foods in human health: a critical appraisal. Am J Clin Nutr 106, 717–724 ,

Inclusion of "milk based drinks" as ultra processed especially needs expansion as milk based beverages can be considered healthy foods if there is minimal added sugar

Line 105 and 153 have a missing word  it now says "carbonated packaged" – carbonated packaged what??

The discussion and conclusions need to focus on all factors including those that were NOT significantly different as those factors are major descriptors of health.  The main takeaway from this study should be that many factors associated with health were not revealed to be differentiated by this analysis of ultra-processed food intake

Author Response

Reviewer 2

  1. Need to detail in abstract factors that were NOT significantly different such as weight, cholesterol levels, waist to hip ratio, metabolic task, waist circumference, depression AND insomnia – emphasizing only minor differences in measures does not accurately reflect the data

Reply: Thank you for this feedback, we have added further detail to the abstract, please see lines 29 – 30.

  1. Definition of "ultra-processed" needs explaining based on the literature not just a list of foods.

Reply: We have added further detail and a reference to the list in the methods section to outline the list was taken from the Australian processed food classification system (lines 105 – 106)

  1. There is debate in the literature that needs to be acknowledge and addressed the only citation is Monteiro et al. The lack of consensus around the NOVA system should be noted see Gibney, MJ, Forde, CG, Mullally, D et al. (2017) Ultra-processed foods in human health: a critical appraisal. Am J Clin Nutr 106, 717–724.

Reply: We have added further detail and references to the limitations section to acknowledge and address this lack of consensus in the literature (Lines 275 – 279).

  1. Inclusion of "milk based drinks" as ultra processed especially needs expansion as milk based beverages can be considered healthy foods if there is minimal added sugar.

Reply: Chocolate milk was considered as milk based beverages. Produced chocolate milk in Iran usually has 22-25 gr sugar per 250 ml. Therefore, chocolate milk was considered as ultra-processed food. Dairy products (no added sugar) were not included in ultra-processed food intake calculation. We have added ‘sugar sweetened’ to ‘milk based drinks’ inline with the NOVA classification to all appropriate references to milk based drinks (please see highlighted areas of the manuscript).  

  1. Line 105 and 153 have a missing word it now says "carbonated packaged" – carbonated packaged what??

Reply: The corrected phrase was carbonated soft drinks and packaged salty snacks. The text was revised please see line: 104 – 105 and 153 – 154.

  1. The discussion and conclusions need to focus on all factors including those that were NOT significantly different as those factors are major descriptors of health. The main takeaway from this study should be that many factors associated with health were not revealed to be differentiated by this analysis of ultra-processed food intake

Reply: Thank you, we have now added this detail to the first paragraph of the discussion and made an amendment to reflect the factors that were not significantly different in the abstract and conclusion. (Please see lines 29 – 30, 202 – 213 and 291 - 294)

Round 2

Reviewer 1 Report

To avoid exaggerating the study theme, the title of “Association between Ultra-Processed Foods with Chance of Insomnia and Quality of Life among Adolescent Girls” should add “in northeastern Iran” at the end.

Regarding Reply 3, does it mean that the FFQs were given to participants throughout a whole year in the study. Then, maybe somewhere in the manuscript, should authors need to address, when did they start giving FFQs, and when the FFQs were done for data analysis.

In Table 2, data of Age, Height, Weight, Waist Circumference, and waist-to-hip ratio look ok, but why does the Percentile BMI have values that high, at least 52 in each quartile. Would authors agree to “participants were all overweight/having obesity”? But, the Height and Weight look fine.

Line 85, m”2”, superscript

Line 87, was a 14 h overnight fast normal? Why was it not 10 or 12 h?

Last, the reviewer would like to preserve his opinions on the hypothesis of the study, even authors answered that similar associations were found in some other countries’ studies. The reviewer is still having doubts about the “Association between the Degree of Processing of Consumed Foods and Sleep Quality in Adolescents” because there are more causes other than eating habits based on sciences.

Author Response

Dear Editor and Reviewers,

Many thanks for taking the time to review the manuscript a 2nd time. We have made further amendments to reflect your useful feedback, highlighted the appropriate amendments to the article in green and noted the line numbers with our comments below.

Reviewer 1

  1. To avoid exaggerating the study theme, the title of “Association between Ultra-Processed Foods with Chance of Insomnia and Quality of Life among Adolescent Girls” should add “in northeastern Iran” at the end.

Reply:

Thank you for the suggestion. We have amended the title to include north eastern Iran.

  1. Regarding Reply 3, does it mean that the FFQs were given to participants throughout a whole year in the study? Then, maybe somewhere in the manuscript, should authors need to address, when did they start giving FFQs, and when the FFQs were done for data analysis.

Reply: FFQ and other questionnaires in our study were filled out at the same time. FFQ allow us to estimate normal food consumption habits over a relatively recent period (months or years) by recording the frequency and amount with which the foods included in a set list are eaten. When the individuals completed FFQ in our study, their dietary intakes were reported during the past year. We should complete FFQ throughout a whole year.

  1. In Table 2, data of Age, Height, Weight, Waist Circumference, and waist-to-hip ratio look ok, but why does the Percentile BMI have values that high, at least 52 in each quartile. Would authors agree to “participants were all overweight/having obesity”? But, the Height and Weight look fine.

 Reply: BMI percentile mean for age is not high for our study. The BMI percentile between percentiles 3 to 85 are considered normal BMI.

  1. Line 85, m”2”, superscript

Reply:

We have corrected this error, now line 86.

  1. Line 87, was a 14 h overnight fast normal? Why was it not 10 or 12 h?

Reply: Thanks for valuable comments. We need to fast for a minimum of 10 hours to a maximum of 14 hours before a test

  1. Last, the reviewer would like to preserve his opinions on the hypothesis of the study, even authors answered that similar associations were found in some other countries’ studies. The reviewer is still having doubts about the “Association between the Degree of Processing of Consumed Foods and Sleep Quality in Adolescents” because there are more causes other than eating habits based on sciences.

Reply:

We thank you for the comment. In defence, our statistical model did adjust for BMI, energy intake, and physical activity. While this does not cover all variables that can influence the quality of sleep, such as psychological issues. However, we feel this would be beyond the scope of this study.  Our limitation section now provides an additional statement on line 271:

‘Finally, sleep quality can be influenced by other factors such as psychological outcomes that were not addressed in our statistical model’

Reviewer 2 Report

The authors have responded to my concerns and improved the manuscript greatly.  

Author Response

Dear Reviewer 2

Many thanks for taking the time to review the manuscript a 2nd time